# Reinforcement Learning Based Fast Self-Recalibrating Decoder for Intracortical Brain–Machine Interface

**DOI:** 10.3390/s20195528

**Published:** 2020-09-27

**Authors:** Peng Zhang, Lianying Chao, Yuting Chen, Xuan Ma, Weihua Wang, Jiping He, Jian Huang, Qiang Li

**Affiliations:** 1Wuhan National Laboratory for Optoelectronics, Huazhong University of Science and Technology, Wuhan 430074, China; hustzp@hust.edu.cn (P.Z.); M201872190@hust.edu.cn (L.C.); chenyuting@hust.edu.cn (Y.C.); 2Department of physiology, Feinberg School of Medicine, Northwestern University, Chicago, IL 60611, USA; xuan.ma1@northwestern.edu; 3School of Artificial Intelligence and Automation, Huazhong University of Science and Technology, Wuhan 430074, China; m201872758@hust.edu.cn (W.W.); huang_jan@mail.hust.edu.cn (J.H.); 4Advanced Innovation Center for Intelligent Robots and Systems, Beijing Institute of Technology, Beijing 100081, China; jiping.he@asu.edu

**Keywords:** intracortical brain–machine interface, reinforcement learning, adaptive decoder, transfer learning

## Abstract

Background: For the nonstationarity of neural recordings in intracortical brain–machine interfaces, daily retraining in a supervised manner is always required to maintain the performance of the decoder. This problem can be improved by using a reinforcement learning (RL) based self-recalibrating decoder. However, quickly exploring new knowledge while maintaining a good performance remains a challenge in RL-based decoders. Methods: To solve this problem, we proposed an attention-gated RL-based algorithm combining transfer learning, mini-batch, and weight updating schemes to accelerate the weight updating and avoid over-fitting. The proposed algorithm was tested on intracortical neural data recorded from two monkeys to decode their reaching positions and grasping gestures. Results: The decoding results showed that our proposed algorithm achieved an approximate 20% increase in classification accuracy compared to that obtained by the non-retrained classifier and even achieved better classification accuracy than the daily retraining classifier. Moreover, compared with a conventional RL method, our algorithm improved the accuracy by approximately 10% and the online weight updating speed by approximately 70 times. Conclusions: This paper proposed a self-recalibrating decoder which achieved a good and robust decoding performance with fast weight updating and might facilitate its application in wearable device and clinical practice.

## 1. Introduction

In intracortical brain–machine interfaces (iBMIs), neural electrodes are chronically implanted into the cortex to record the neural activity, which is then translated into control commands on assistive devices for helping amputees or paralyzed patients restore their motor functions [1,2]. With rapid development over past decades [3,4,5,6,7,8], iBMIs have achieved significant improvements and can assist paralyzed patients to control their artificial limbs while performing three-dimensional movements [6,9] or tapping on a screen to communicate with others [8,10,11,12]. However, the decoder, which is an important part of an iBMI, needs daily retraining in a supervised manner to maintain a robust decoding performance [13,14]. The decoder is referring to a general set of algorithms commonly used in iBMI decoding.

Daily retraining of the decoder has to be implemented for the non-stationarity of the neural recordings [15,16]. Owing to a micromotion or insulation degradation of the implanted electrode, a biological foreign body response, and the physiological characteristics of the neurons, neural recordings are non-stationary [17,18,19], which causes variation over time in the relationship between the neural recordings and the specific movement intentions. Therefore, the decoder of the iBMIs must be recalibrated before usage for proper application [20,21]. To avoid recalibration of the decoder, a direct approach is to design a more powerful decoder that can handle the nonstationarity of the neural recordings. With the development of deep learning [22,23], this approach can be realized. Deep neural network has achieved good and very robust performance [19] but requires large amounts of samples to train the model. An alternative to daily retraining is to employ an adaptive neural decoder that can automatically update the parameters and track the dynamic neural activity [21,24]. Some studies have implemented this method and have achieved good decoding performance; however, most of them commonly employ supervised learning and train the decoder by mapping the recorded neural activities to some kinematic outputs, such as the real movement trajectory or the movement labels [25,26,27]. In clinical applications, the kinematic outputs may be difficult to collect, particularly for paralysis or limb amputations [28,29,30,31]. To address this problem, reinforcement learning (RL)-based iBMIs have been developed. RL is a machine learning method that enables an agent to learn which action obtains the best reward in an interactive environment through trial and error [32]. Different from supervised learning, RL used a scalar reward for positive behavior after tasks [33]. In the RL based iBMIs, the user is the agent, and the iBMI system is the environment. The RL-based iBMI allows iBMI users to modify their brain activities and learn from their interaction with the environment [34]. Simultaneously, a reward signal, instead of limb kinematics, sent from the environment is used to reinforce the decoder according to the accomplishment of the task without the real movements of the patients. Therefore, an adaptive decoder based on RL may be a good option for iBMIs in clinical applications. 

Researchers have proposed some adaptive RL algorithms for iBMIs. DiGiovanna et al. [35] first introduced a paradigm in which a rat brain controlled a robotic arm to discriminate between two targets. A neural network was used to represent non-linear state-action mapping. A temporal difference error was employed to update the Q value for each action. Sanchez et al. [36] proposed a novel RL-based BMI (RLBMI) architecture and tested three rats in controlling a robotic arm in performing a two-target choice task. They then used Hebbian reinforcement learning to design an adaptive controller for use by marmosets, which achieved a good performance in controlling the reaching movements of a robot arm [37]. They further tested the performance of the RLBMI in facing the dramatic perturbations of the neural inputs and still achieved a robust performance [28]. Moreover, Sanchez et al. [38,39] proved that the reward information in the autonomous RLBMI can be represented by the neural activities from the nucleus accumbens, and Marsh et al. [40] further showed that the reward information can also be represented by the neural activities from the primary motor cortex. The above studies are mainly based on Q-learning, which tends to cause the curse of dimensionality and is inefficient in exploring new knowledge. To increase the exploration efficiency of the RLBMI, Wang et al. [41] introduced attention-gated reinforcement learning (AGREL) to decode more complicated actions and achieved a better decoding performance than the Q-learning method [34]. These studies have achieved good results, and AGREL showed significant potential in iBMIs applications. However, AGREL is sensitive to initialization and has to re-initialize numerous times to avoid becoming stuck in a poor performance, which affects the real-time capabilities and prohibits it from meeting the requirements of online decoding. Moreover, disparities between historical and new data caused by nonstationarity enlarge the state-action space and bring about more challenges to the exploration efficiency [29,30].

Motivated by the above problems, in this study, transfer learning (TL) is introduced and integrated with RLBMI. TL can extract knowledge from the source tasks for application to a target task [42]. Therefore, knowledge extracted from the historical data can be used to prime the RL decoder with current data [15]. The learning task for RLBMI is then easier, and the efficiency can be improved. To our best knowledge, this is the first time that TL is introduced to the RLBMI. This may be due to the fact that the RLBMI is applied in the online decoding situation, therefore, no target domain data and labels can be used, which makes it difficult to implement the conventional TL method. The TL method used in this study extracts the projected feature space only based on the training data in the unsupervised manner and can be well integrated with the RLBMI. In addition, mini-batch (MB) and some other weight updating schemes are integrated to further improve the performance of the RLBMI. Generally, we proposed a transfer learning and mini-batch based attention-gated reinforcement learning algorithm, which was named as TMAGRL. In the proposed TMAGRL algorithm, we first implemented a principal component analysis-based domain adaptation (PDA) method to project both the training data and the testing data into the same feature space, which was extracted from the training data, thus diminishing the disparities between them while decreasing their dimensions. We then updated the model using an MB method for selecting a small number of the latest current samples instead of using all training data; we could accelerate the model updating and help mitigate against over-fitting. Finally, we proposed certain weight updating schemes to optimize the updating procedure and further avoided over-fitting. The TMAGRL algorithm was tested on real intracortical neural data recorded from two monkeys performing reaching and grasping tasks and was compared with other decoder calibration methods.

The main contributions of this paper are as follows: (1) We proposed a new RLBMI algorithm, TMAGRL, which can overcome the difficulty in combining general TL with the online-RLBMI by extracting the projected feature space only from the source domain in an unsupervised manner and solves the problems of low learning efficiency and unstable performance in conventional RLBMI. This might be the first time TL has been integrated with the RLBMI. (2) We also introduced MB and weight updating schemes to the RLBMI to further speed up the weight updating and help mitigate over-fitting. (3) The performance of TMAGRL was tested on real intracortical neural data recorded from two monkeys performing different behavioral paradigms. TMAGRL achieved a superior performance on the data of both monkeys, indicating a viable generalization. (4) Higher decoding accuracy and faster online weight updating were achieved by TMAGRL compared with the other decoding methods, which might facilitate the application of the RLBMI in wearable device and clinical practice.

The rest of this paper is organized as follows: Section 2 describes the procedures of the animal experiments and the preparation of the neural data. The adaptive decoder based on TMAGRL is then presented. Section 3 details the decoding results of the comparison between TMAGRL and the other methods. This is followed by the results of the parameter updating efficiency in TMAGRL and the effect of the batch size on the TMAGRL performance. We then discuss the experimental results in Section 4. Finally, Section 5 provides some concluding remarks.

## 2. Materials and Methods 

All experiments and surgical procedures used in this study were approved by the Institutional Animal Care and Use Committee at Huazhong University of Science and Technology.

### 2.1. Experimental Setup and Neural Recording

The experiment was conducted with two adult male rhesus macaques (M and B), which were trained to perform spatial reaching and grasping movements. The monkeys were seated in a custom primate chair with the left arm restricted, and the right hand was used to reach and grasp the target object on the experimental apparatus. The experimental apparatus mainly contained a center pad and three target objects, as shown in Figure 1, the details of which are described in [43]. For monkey M, all three target objects were cubes. The monkey was guided to reach and grasp the target objects in three different positions. Unlike monkey M, these three target objects were of different shapes (cube, triangle, and sphere) for monkey B. Monkey B was guided to reach the same position and grasp the target object of different shapes that could be transferred to the same position using the turntable of the apparatus. A trial began with the center light on, which guided the monkey to touch the center pad. Following a hold time of approximately 500 ms, the target light was on and guided the monkey to reach and grasp the target object. After the monkey grasped the target object and held it for approximately 300 ms, the target light went off, and the monkey would receive a liquid reward.

When the monkeys became familiar with the experiment, microelectrode arrays were implanted into the cortex using standard neurosurgical techniques. We implanted the electrode arrays into the arm/hand area of the primary motor cortex (M1), the somatosensory cortex (S1), and the posterior parietal cortex (PPC), as identified by the local anatomical landmarks and further confirmed through an intracortical microstimulation [15]. The electrode locations are shown in Figure 1. For monkey M, a 32 channel Utah array was implanted in M1 and S1, respectively, and a 16 channel FMA array was implanted in the PPC.array was implanted in the PPC. Four 32 channel FMA arrays were implanted in monkey B, two in S1, and the other two in M1 and PPC. Neural activities and behavioral data were recorded using a 128 channel Omniplex system (Plexon, Inc.) with a sample rate of 40 kHz. The wideband signals were band-pass filtered between 250 Hz and 6 kHz. Threshold crossing [44] was used to detect the spike, and the threshold was set as a value of –4.5 times the root mean square of the signal in each channel. All channels of the arrays in monkey M worked well, whereas only 66 channels of the arrays in S1 and PPC could record spike signals for monkey B. The recordings from these working channels were used in the following analysis.

### 2.2. Feature Extraction and Data Preparation

In this study, monkey M was trained to reach three different positions; therefore, a 300 ms time window with 200 ms before and 100 ms after the center release event was chosen to extract features with 50 ms bins, and the spike counts in each bin were used as the feature. Monkey B was trained to grasp three different shapes of the target objects; therefore, a 300 ms window with 100 ms before and 200 ms after the target hit event was chosen to extract the features with 50 ms bins. For both monkeys, a 300 ms window was chosen for decoding, and six features could be extracted for each channel. The details of the feature extraction are shown in Figure 2.

The experimental data used in this study were collected during a period of approximately one month. The data from monkey M consisted of three sessions, and the datasets in each session were collected from consecutive days during the same week. The first session contained four datasets, and the other two sessions contained five datasets. There were also three data sessions for monkey B, each of which contained four consecutive datasets. The datasets in the same session were consecutively collected for several days and labeled according to the name of the session; for example, “S1D1” meant the dataset was from the first day of session 1. Therefore, if S1D4 was selected as the current data, S1D3, S1D2, and S1D1 all could be regarded as the historical data. Each dataset contained approximately 600 successful trials for monkey M and approximately 300 successful trials for monkey B.

### 2.3. Transfer Learning and Mini-Batch Based Attention-Gated Reinforcement Learning

#### 2.3.1. Neural Network Structure of TMAGRL

The neural network structure of TMAGRL was based on AGREL [41], and it adopted a simple three-layer network structure to translate neural activities into action states. The output was evaluated using an instantaneous reward [34], as shown in Figure 3. The neural network synaptic weights were updated according to a simple and physiologically plausible Hebbian rule [42]. The neural activity vector (NAV) of monkey M had 480 feature elements (80 working channels with 6 features for each channel). The NAV of monkey B had 396 feature elements (66 working channels with 6 features for each channel). Therefore, the numbers of input neurons of the network were 480 and 396, respectively. To map the neural activities to three action states, the output layer of the neural network had three output units. The number of units in the hidden layer of the neural network was 30 (M = 30). The weights of the neural network, $v_{ij}$ and $w_{jk}$, were randomly initialized at between ±0.1, and the learning rate was set to 0.01 [34].

The decoder hidden layer used a sigmoid nonlinear activation function. Connections $v_{ij}$ propagated the activity from the input layer to the hidden layer. The output of the hidden layer is written as follows:(1)$$\gamma_{j}=\frac{1}{1+\exp(-h_{j})}\;with\;h_{j}=\sum_{i=0}^{N}v_{ij}x_{i}$$

Connections $w_{jk}$ propagated the activity from the hidden layer to the output layer. After the output layer, the decoder adopted a stochastic softmax rule to calculate the probability that each output unit was selected. During every trial, only the winning unit achieved activity 1, whereas the other output units were set as activity 0. The probability of the winning unit is defined as follows:(2)$$P(Z_{k}=1)=\frac{\exp(a_{k})}{\sum_{k^{\prime }=1}^{C}\exp(a_{k^{\prime }})}\;with\;a_{k}=\sum_{j=0}^{M}w_{jk}\gamma_{j}$$

If the decoder chose the correct action, the network received a reward ***r***, and we assumed that *r* equaled 1. If the decoder chose an incorrect action, the network was not rewarded. During the rewarded trials, a global error signal δ was defined as follows: (3)δ=r−E(r)

Here, δ equals the difference between the amount of reward obtained and the amount expected for a particular trial. In unrewarded trials, δ equals −1, which was used to give negative feedback to the network. Finally, an expansion function was used to determine the change in network weight.
(4)$$f(\delta )=\{\begin{array}{c} \frac{\delta }{1-\delta },\;\delta \geq 0\\ \delta ,\;\delta =-1 \end{array}$$

Here, f(δ) is a biologically plausible expansive function. Specifically, if the decoder chose the correct output action with a lower probability, it gave a stronger positive feedback to the network. After each trial, the synaptic weights were updated according to a simple and physiologically plausible Hebbian rule. In addition, δ determined the weights between the input units and hidden units by the expansive function.
(5)$$\Delta w_{jk}=\beta \gamma_{j}Z_{k}f(\delta )$$

The weights $v_{ij}$ between the input layer and the hidden layer were also modified according to the Hebbian rule, which depended on f(δ). Here, $w_{js}$ represents the feedback of the winning unit s.
(6)$$\Delta v_{ij}=\beta x_{i}\gamma_{j}f(\delta )[w_{js}(1-\gamma_{j})]$$

Because the output units participated in the competition, the winning unit received activity 1, and the other units received activity 0; therefore, only the weight of the current selected output action was updated. Hidden units that provided the highest excitation to the winning output unit also received the strongest feedback. The feedback thus assigned credit to the hidden units responsible for the choice of action, which is called the attention mechanism.

#### 2.3.2. Training Algorithm for the Initial Decoder of TMAGRL

With TMAGRL, the basic structure described above was combined with the TL, the MB, and the weight updating schemes for the decoder construction and updating. During the online decoding of one dataset, only the historical data could be used for constructing the initial decoder, the current data were invisible, and each current sample appeared in turn for testing. Because of the nonstationarity of the neural recordings, there were disparities between the historical data and the current data. The decoder directly calibrated by the historical data could not decode the current data accurately without a decoder adaptation, and the disparities increased the difficulty in the decoder adaptation. Therefore, TL was introduced to diminish the disparity for accelerating decoder adaptation. We used the PDA algorithm to project the historical data and current data to a new feature space to diminish the disparity [15]. In the PDA algorithm, a principal component analysis (PCA) was applied to the historical data, and the d eigenvectors corresponding to the d largest eigenvalues were used as the new feature space $M_{s}$. The value of d was chosen by retaining 90% of the original data variance in this study. The $M_{s}$ projection is described as follows:(7)$$Z_{s}=S_{s}\times M_{s}$$
where $S_{s}$ is the historical sample, and $Z_{s}$ is the projected data. For the new current data T, each sample $T_{t}$ was projected to this same feature space $M_{s}$ before testing.
(8)$$Z_{test}=T_{t}\times M_{s}$$

After applying the PDA, the decoder was trained using the projected historical data. During the decoder training process, our goal was to ensure that more samples selected the correct action with a higher probability. Therefore, we set up a probability threshold k—if the sample can select the correct action with a probability of more than k, the corresponding weights of the synaptic connections are no longer updated and are applied directly to the next sample forecast. The experiments showed that k = 0.9 was a good choice. As the convergence condition, if 98% of the training samples could choose the correct action with a probability of more than 0.9, then the model was convergent. At the same time, we set the maximum number of epochs to limit the maximum training time. If the maximum number of epochs was reached, but the model could not reach the convergence condition, we selected the optimal trained model. The optimal model was the one that can accurately predict most of the neural signal data during the training process. The process of the decoder training is shown in the flow chart of Figure 4.

#### 2.3.3. Adaptive weight Updating in TMAGRL for Online Testing

During the online testing, after each new sample was tested, the weights of the decoder were updated using the MB sample set. We selected the latest J samples to form the MB sample set. At the beginning of the online testing, the number of tested samples was less than J, and all tested samples were used for weight updating. For the weights between the hidden units and the output units, they were updated using Equation (9), in which $\delta^{n}$ represents the modulation of Hebbian plasticity by the n_th_ sample in the MB sample set.
(9)$$\Delta w_{jk}=\beta \sum_{n=1}^{N_{RTMB}}\gamma_{j}^{n}Z_{k}^{n}f(\delta^{n})$$
(10)$$\Delta v_{ij}=\beta \sum_{n=1}^{N_{RTMB}}x_{i}^{n}\gamma_{j}^{n}f(\delta^{n})fb_{\gamma_{j}}^{n}\;with\;fb_{\gamma_{j}}^{n}=w_{js}^{n}(1-\gamma_{j}^{n})$$

The weights between the input layer and the hidden layer were updated using Equation (10). The attention factor, $fb_{\gamma j}^{n}$, which equaled the output action of the n_th_ testing sample to the feedback of the *j_th_* hidden layer unit, also influenced the plasticity. In addition, $w_{js}^{n}$ indicates the feedback of the winning units s of the *n_th_* sample (red connections in Figure 3).

After finishing the new testing sample prediction, this sample replaced the oldest sample in the MB sample set. We used certain schemes in updating the weight. As with the initial decoder training, if the decoder could choose the correct action with a probability of more than k for the current sample, the decoder would not be updated for this sample. Otherwise, all samples in the MB sample set were used to update the weights of the decoder. The convergence condition was the same as in the initial decoder training. The process of weight updating is shown in the flow chart of Figure 5.

### 2.4. The Evaluation of Weight Updating Efficiency in TMAGRL Method

The weight updating efficiency is an important index for evaluating an adaptive method. The weight updating in the TMAGRL method could be divided into two parts. The first part was the weight updating during the initial decoder training using the historical data, as described in Figure 4. The other part was the weight updating after finishing the sample testing, as described in Figure 5. To test the weight updating efficiency in the first part, the decoder was set to be trained for 1000 epochs, and the weights of the decoder were updated using all historical data during each epoch. The root mean square (RMS) value between the predicted and the actual values was used to describe the training efficiency. This procedure was repeated 50 times to achieve the mean values with a standard deviation. The second part of testing the efficiency of the weight updating took place during the online testing, in which the weights were updated using the new current samples. In this part, we used the weight updating time to evaluate the weight updating efficiency. The time spent in parameter updating during the online testing is extremely important and must be sufficiently short to meet the clinical requirements. The testing results are shown in the Results section.

### 2.5. Other Decoder Calibration Schemes

To comprehensively evaluate the performance of the TMAGRL, four common methods were presented and compared:

1. Static decoder: With a static decoder, the historical data were used for decoder training, and current new samples were tested without an adaptation using the support vector machine (SVM) [24].

2. Retrained decoder: For the retrained decoder, three-fold cross-validation was used with SVM to achieve the decoding accuracy. This scheme was generally applied in offline mode and represented the ideal performance that online decoding could achieve [24].

3. AGREL: AGREL has the same structure as TMAGRL, with the exception of the TL, the MB, and the weight updating schemes, the details of which are described in [34,42].

4. Non-adaptative AGREL (NAGREL): NAGREL has the same structure as AGREL but without an adaptation, and it was presented for a comparison of the effect of the adaptation in AGREL.

To standardize the performance evaluation of these schemes, except for the static decoder, the testing data were tested 50 times to achieve the mean decoding accuracy. Bootstrapping was then used to obtain 95% confidence intervals of the decoding accuracies. In the retrained decoder, the training data were randomly selected from all current data, and the remaining data were tested. This was also repeated 50 times to obtain the mean decoding accuracy. The SVM was implemented based on the LIBSVM, which is a Library for Support Vector Machines [45], and the radial basis function (RBF) kernel was used. Moreover, the grid-search method introduced by the LIBSVM was used to obtain the best penalty and kernel parameters C and γ, respectively, which were the two main parameters for the RBF kernel.

## 3. Results

### 3.1. Performance of TMAGRL in Calibration with Historical Data from the Previous Day

To solve the problem of daily retraining, only historical data could be used for the decoder calibration before using the iBMIs. Under this scenario, the data from the previous day were used to calibrate the decoder to test the new data. For the static decoder, AGREL, NAGREL, and TMAGRL, there were 630 training samples from the previous day and 630 testing samples of the current day in monkey M, while there were 360 training samples from the previous day and 360 testing samples of the current day in monkey B. For the retrained decoder, there were 420 training samples and 210 testing samples from current day in monkey M as well as 240 training samples and 120 testing samples from current day in monkey B. The decoding performances of the presented methods are shown in Figure 6, and the mean decoding accuracies and standard deviation across all data sessions are listed in Table 1. On almost all testing datasets, the TMAGRL method outperformed the other four methods (even the retrained decoder) in the confidence intervals for both monkeys. Moreover, the confusion matrix of decoding results for both monkeys is shown in Table 2. The results in Table 2 are the mean values across all datasets.

SVM is one of the most common and effective classification methods, but the decoding results by the static decoder were not robust, which might have been influenced by the nonstationarity of the neural recordings. For the retrained decoder, it also was unable to achieve a decoding accuracy as high as TMAGRL (monkey M: F(1, 9)= 4.72, p = 0.04; monkey B: F(1, 7)= 7.19, p = 0.02, ANOVA test), which might have been caused by the nonstationarity among the samples of the current day. Similar to the static decoder, NAGREL was also nonadaptive, and the decoding performance was not robust. The AGREL method, which updated parameters adaptively, achieved a similar decoding accuracy as the NAGREL method that used constant parameters. With improvements, TMAGRL achieved a higher and more robust decoding performance than AGREL in both monkeys.

### 3.2. Performance of TMAGRL in Calibration using Historical Data with Higher Time Separation

Normally, the disparities between the two datasets increased with an increase in the time separation. Greater disparities resulted in more difficulties in the adaptation of the decoder. In the above analysis, the TMAGRL method achieved an effective decoding performance using historical data from the previous day to initially train the decoder. Here, we tested the decoding performance of the TMAGRL method using historical data with a greater time separation. For each data session of both monkeys, the last dataset in this session was used as the testing data, and data from the other datasets were separately used for training. Therefore, the decoding performance of the decoder using historical data with 1, 2, 3, and 4 day separations were compared, the results of which are shown in Figure 7. The mean decoding accuracies and standard deviations across all data sessions are listed in Table 3.

In general, the results in Figure 7 were consistent with those in Figure 6, and the TMAGRL method outperformed the other four methods. For the static decoder and the nonadaptive NAGREL method, a greater time separation between the training and the testing data achieved a lower decoding accuracy, which indicated that the disparities between the two datasets increased with the increase of the time separation. However, the decoding accuracies of the TMAGRL method did not follow this rule and were unaffected by the increased disparities. There was no significant difference in decoding results of the experiments using historical data with different time separations (monkey M: p > 0.05; monkey B: p > 0.05 ANOVA test). Because the adaptive capabilities of TMAGRL have an important role, the adaptive decoder has a strong ability to correct its own parameters according to the disparities between the training and the current sample sets. The decoding performance of TMAGRL was unaffected by the increased time separation in the same session, which can improve the practicality of this method in clinical applications. By contrast, the decoding results of the AGREL method were still unstable.

### 3.3. Weight Updating Efficiency in TMAGRL Method

As described in Section 2.4, the evaluation of weight updating efficiency in the TMAGRL method contained two parts. For the first part, the first dataset in session one of both monkey M and monkey B was taken as the example, and the results of TMAGRL and AGREL are shown in Figure 8. It can be seen that the curve of the TMAGRL method was much smoother than that of AGREL. TMAGRL began to converge at approximately 200 epochs, whereas AGREL began to converge at more than approximately 600 epochs in monkey B. The convergence speed of TMAGRL was faster than that of AGREL. Moreover, the standard deviation of the RMS for the AGREL method was much larger than that of the TMAGRL method, which indicated that TMAGRL converged more easily than AGREL and that the convergence of the AGREL was not robust. In this part, the difference between TMAGRL and AGREL was that the TMAGRL used PDA to project the historical data into a new feature space. Therefore, the improvement in the initial decoder training was achieved by the PDA in TMAGRL.

For the second part, the mean weight updating time after each sample testing in the AGREL and TMAGRL methods, in which the initial decoder was trained using historical data from the previous day, are summarized in Table 4. For monkey M, AGREL took approximately 391 ms to finish the weight updating, whereas TMAGRL took only approximately 5 ms. For monkey B, the weight updating times were 298 ms and 4 ms for the AGREL and the TMAGRL methods, respectively. The updating speeds of the TMAGRL method were approximately 77 and 76 times faster than those of the AGREL method, which was critical for iBMIs used in clinical applications. 

### 3.4. Effect of Batch Size on Performance of TMAGRL

In this study, the MB was introduced in the online weight updating of TMAGRL to reduce the amount of data used for updating while increasing the updating speed. Differing from the full batch that used all new current data to update the weights in each epoch, the MB only used the latest J current samples for updating the weight. The choice of batch size J influenced the decoding performance of the decoder. A smaller batch size could accelerate the weight updating but might reduce the decoding accuracy, whereas larger batch sizes might achieve a better decoding accuracy but took more weight updating time and affected the real-time performance. We subsequently explored the impact of batch size J on the performance of TMAGRL. With batch size J changing from 30 to all sample sizes applied with steps of 30, the corresponding decoding accuracy of the TMAGRL was calculated as shown in Figure 9. In this figure, a batch size of one, which used only the latest single sample for updating, was also presented for comparison.

The results indicated that the decoding performance when using a batch size of one was not as robust as that using a larger batch size. A batch size of 30 and above achieved a good and robust decoding accuracy. The weight updating time did not increase with the batch size, which might have been caused by our proposed weight updating schemes. With our updating schemes, if the decoder could make the right prediction with a probability of more than the threshold in new sample testing, the decoder would not be updated for this sample. Therefore, a decoder that was updated with a bigger batch size could maintain more robust decoding performance and might not be updated as frequently as that with a smaller batch size. In other words, the update speed of a small batch size was fast, but the update times were longer, while the update speed of a large batch size was slow, but the update times were shorter. However, the user would not want to spend much time on decoder updating in a single trial. Therefore, a small batch size (such as 30) might be a better choice.

## 4. Discussion

In this study, we proposed an online self-recalibrating decoder, TMAGRL, which combined RL with TL. Using the RL, the decoder was reinforced using only a scalar evaluative signal (reward), and the data labeling was not necessary, which improved the practicability of the iBMIs. Using TL, the disparities between the historical data and the current data could be diminished, which reduced the difficulty in the decoder adaptation. By further integrating the MB and the weight updating scheme, TMAGRL achieved approximately 90% classification accuracy in all datasets for both monkeys, and the weight updating speed increased by more than 70 times.

To evaluate the TMAGRL method comprehensively, AGREL and other related methods were presented for comparison. Based on the results of the non-adaptive methods, we found that nonstationarity was widespread between the historical and the current data. Without an adaptation, the decoding performance decreased for this nonstationarity. Moreover, longer time separation between the historical and the current data could have allowed greater nonstationarity to occur and worse decoding performance to be achieved. The static decoder was nonadaptive and could achieve good decoding results only when the data from the current day were similar to the data from the previous day. The retrained decoder also was unable to achieve a decoding accuracy as high as TMAGRL, because it could not solve the nonstationarity among the samples of the current day, whereas TMAGRL solved it through adaptation. NAGREL was also nonadaptive, and the decoding performance was not robust. However, NAGREL achieved higher decoding accuracies than the SVM, which indicated that the basic AGREL was an effective classification method but was not good at solving the nonstationarity of the data from different days. 

Compared to the non-adaptive methods, the adaptive decoder can solve the problem of this nonstationarity. However, the decoding performance of AGREL was not robust because it was sensitive to an initialization and could become stuck in the local minima, particularly for online learning [34]. This instability of AGREL would be further exacerbated when the disparity between historical and current data increased. Compared with AGREL, TMAGRL used TL (PDA algorithm) to diminish the disparities between historical and current data while also eliminating some of the noise and reducing the computational burden. Therefore, TMAGRL converged faster and accelerated the weight updating, and thus the performance instability of AGREL was effectively overcome.

In addition to TL, we also proposed the use of an MB and a weight-updating scheme in TMAGRL. AGREL used all new current data to update the weights after each trial testing, which led to an increased amount of updated data to affect the weight updating speed. The MB could effectively solve this problem by using only the latest J samples for a weight updating. The results in Figure 9 indicated that the decoding accuracies were unaffected compared to the full batch updating. However, we wanted to point out that using the latest J samples to form the MB might not be the best solution. Compared to our approach which belongs to the experience replay, the prioritized experience replay might further improve the decoding performance [46]. Even though our MB scheme was not the best solution, it was simple and did not need to spend time in choosing suitable samples. Therefore, the real-time performance of our MB scheme was good, which was important for online decoding. Moreover, we used a weight-updating scheme in TMAGRL to avoid the over-fitting problem. In AGREL, Equation (4) gave unexpected rewards with more weight updating, which might prevent over-fitting. However, this cannot stop samples with a high prediction accuracy from participating in the weight updating, resulting in over-fitting. In our weight updating scheme, samples with a prediction accuracy above the threshold cannot participate in the updating, which could effectively prevent over-fitting while reducing the burden on the weight updating and improving the updating speed.

In this study, we integrated the TL with RL technology to solve the decoder recalibration problem, which was caused by nonstationarity of neural recordings. Nonstationarity of neural recordings also exists in electroencephalogram (EEG)-based BMI. TL based methods have been used to solve this problem. Krauledat et al. proposed a TL approach on common spatial patterns to finish the session-to-session transfer by finding invariant common spaces to project the new testing data [47]. Using the same idea that knowledge is transferred by finding invariant common spaces, this work was extended to subject-to-subject transfer by many other researchers [48,49,50]. Meanwhile, Sam et al. proposed a TL approach of stationary subspace analysis and attempted to find a stationary subspace of data from multiple subjects and/or sessions [51]. Many TL methods have been applied in the EEG-based BMI and achieved good decoding performance. However, these methods were not integrated with RL for online decoding. During RL-based online decoding, no target domain data and labels could be used, which made it difficult to implement a conventional TL method. Our TMAGRL overcame this difficulty and might be used in EEG-based BMI research to achieve better decoding performance.

In general, TMAGRL achieved a higher decoding accuracy and a faster decoding speed than AGREL. However, in this study, TMAGRL was tested using the collected data, not in the total online brain control task. Therefore, only the adaptive ability of the decoder was tested. For the brain control task, the brain, the decoder, and the environment are considered as a whole, and the brain is continuously adapting while the decoder updates the weights, which is the process of co-adaptation. Moreover, this study only used the TMAGRL for three classifications; the study about using TMAGRL for more classifications or continuous neural decoding can be explored. Additionally, learning from other RL system [52] or using a wearable sensor system [53,54] to determine the reward of the RL system may further improve the practicability. In the future, we intend to test our method integrated with a wireless wearable sensor system on a brain control task to further verify its effectiveness in clinical application.

## 5. Conclusions

In this paper, an adaptive decoder based on the TMAGRL algorithm was proposed to solve the problem of daily retraining in a supervised manner. The TMAGRL algorithm combined transfer learning with reinforcement learning to construct an adaptive decoder. This adaptive decoder can effectively close the gap between historical and current data, and it achieved a good and robust decoding performance. The adaptive decoder did not require an every-day recalibration; only the neural data after each testing were needed in the minibatch calibration to update the weights of the decoder in real time. Moreover, this adaptive decoder could maintain a good decoding performance even when the historical data used for the initial training were obtained from some of the previous days. In addition, the weight updating speed was improved by approximately 70 times when using TMAGRL compared to the original AGREL, which might improve the practicability of iBMIs in clinical applications.

## Figures and Tables

**Figure 1 sensors-20-05528-f001:**
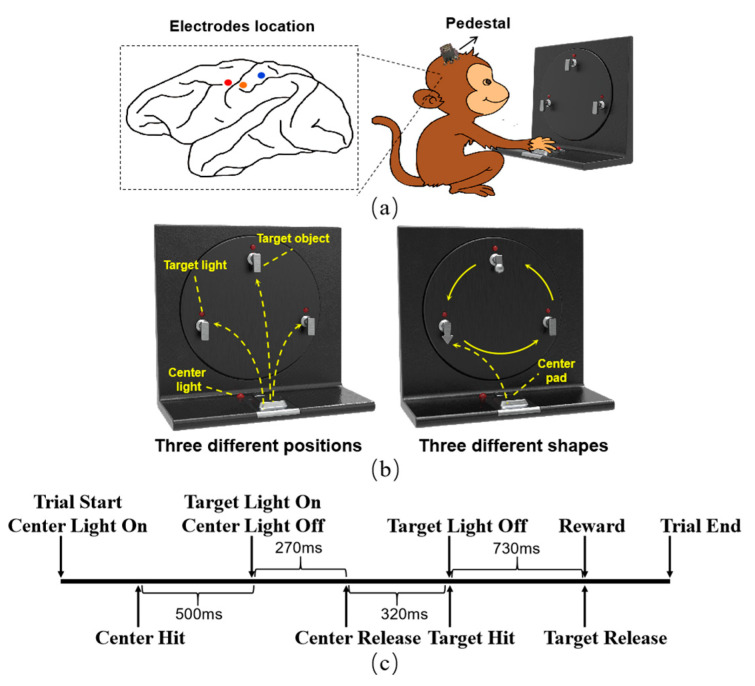
Electrode location, experimental apparatus, and sequences of the experimental paradigm. (**a**) The electrode location. Electrode arrays were implanted into the primary motor cortex (M1) (**red circle**), the somatosensory cortex (S1) (**orange circle**), and the posterior parietal cortex (PPC) (**blue circle**) cortex, and the interfaces of the arrays were fixed on the skull with a pedestal. (**b**) Experimental apparatus. Monkey M was guided to grasp the target objects with the same shape at three different positions. Monkey B was guided to grasp three target objects with different shapes at the same position. (**c**) Sequence of the experimental paradigm. The time for the monkeys to perform these actions was different for each trial; the timings were mean value across all the trials of two monkeys.

**Figure 2 sensors-20-05528-f002:**
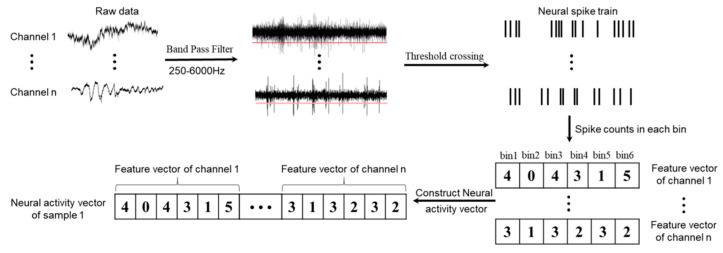
For one sample, the raw data in each channel were band-pass filtered between 250 Hz and 6 kHz and were then detected using the threshold crossing method to obtain neural spike trains, and the threshold was set as a value of –4.5 times the root mean square of the signal in each channel. A 300 ms decoding window with six bins was chosen, and the spike counts in each bin were used as the feature, and thus each channel had six features. The feature vectors of all channels were then combined together to construct the neural activity vector, which had n × 6 features.

**Figure 3 sensors-20-05528-f003:**
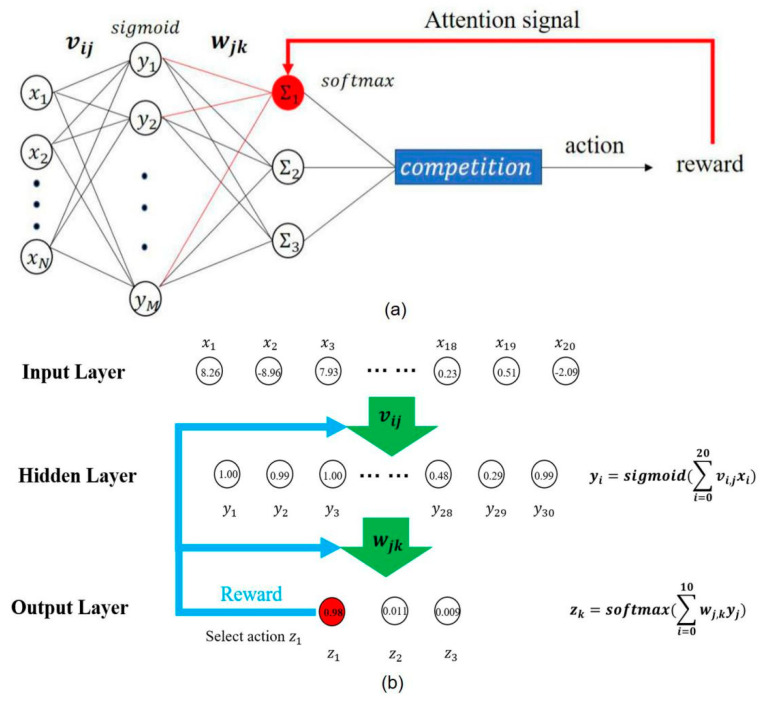
(**a**) Neural network structure of transfer learning and mini-batch attention-gated reinforcement learning (TMAGRL). Three-layer neural network was trained to perform a mapping task of the neural activities to the output states. The winning unit (the unit marked in red) fed its activity back to the hidden layer through the connections (**red lines**) attached to it. (**b**) An example of the feature representation learned by neural network. Connections $v_{ij}$ propagated activity from the input layer to the hidden layer, and connections $w_{jk}$ in turn propagated the activity from the hidden layer to the output layer.

**Figure 4 sensors-20-05528-f004:**
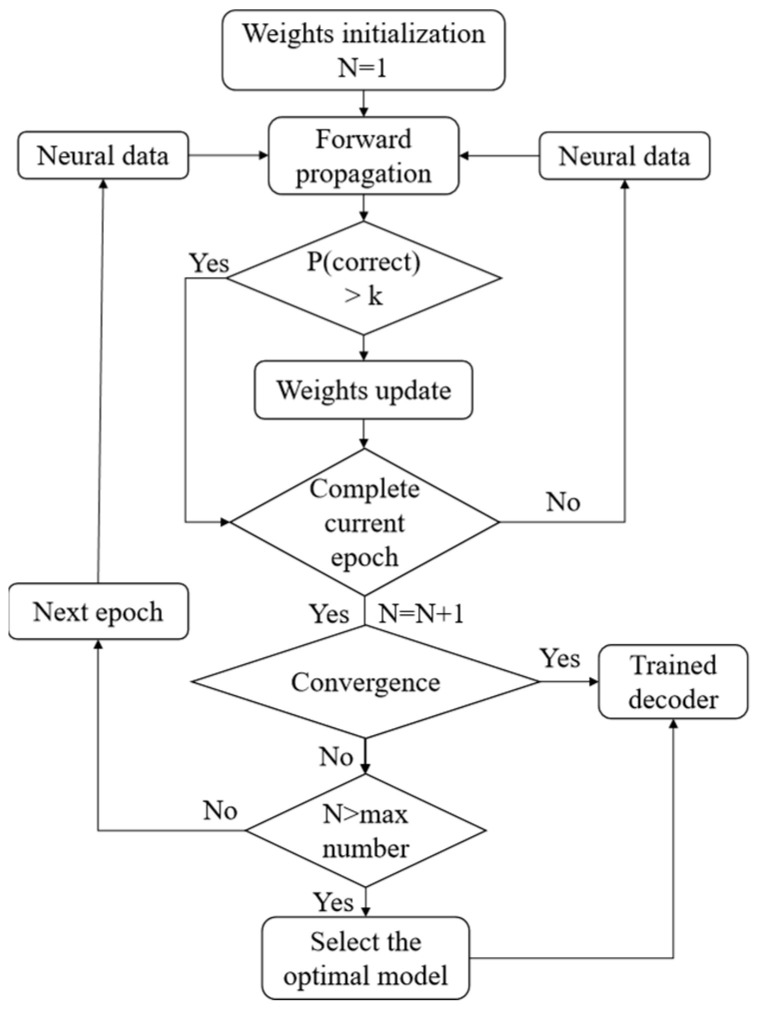
Flow chart of initial decoder training.

**Figure 5 sensors-20-05528-f005:**
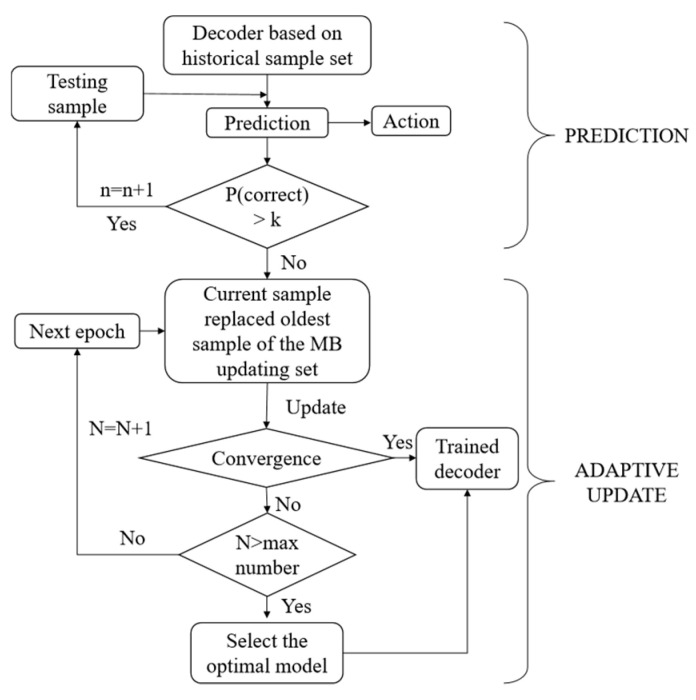
Flow chart of adaptive weight updating.

**Figure 6 sensors-20-05528-f006:**
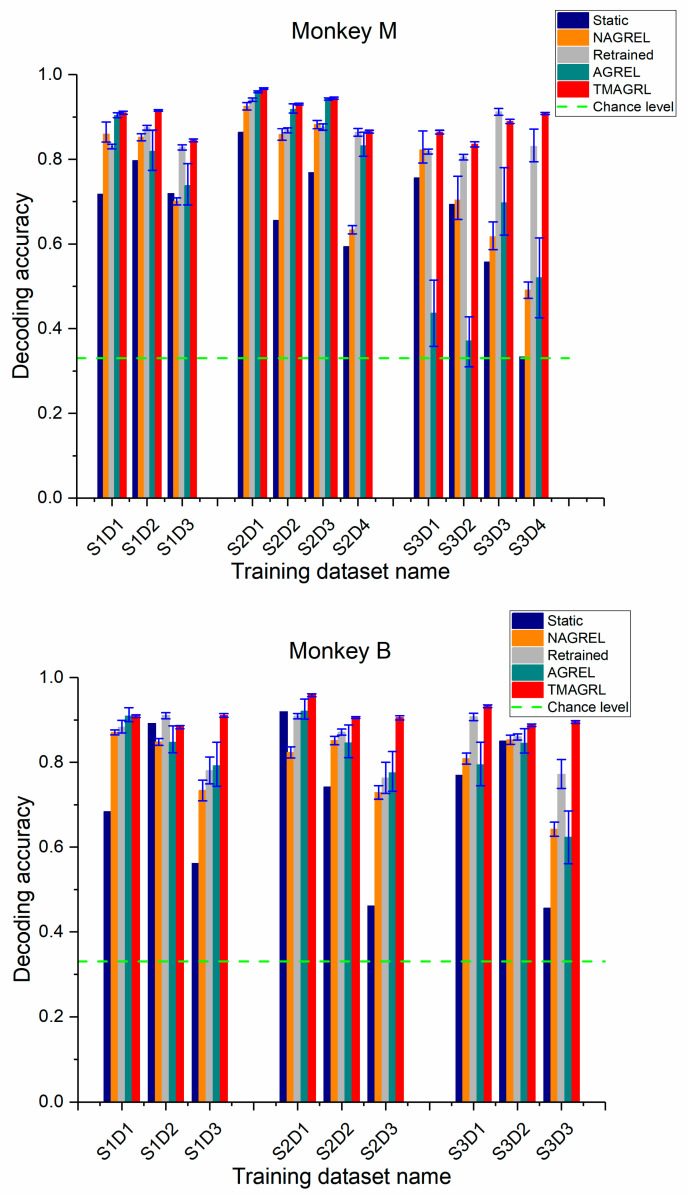
Performance comparison of five methods across all sessions for both monkeys. The decoding accuracy of each dataset was the mean value of 50 repeated computations. Blue, orange, gray, green, and red bars represent the classification accuracy of the decoders calibrated using a static decoder, NAGREL, retrained decoder, AGREL, and the proposed TMAGRL, respectively. Error bars represent the confidence intervals of the mean decoding accuracies.

**Figure 7 sensors-20-05528-f007:**
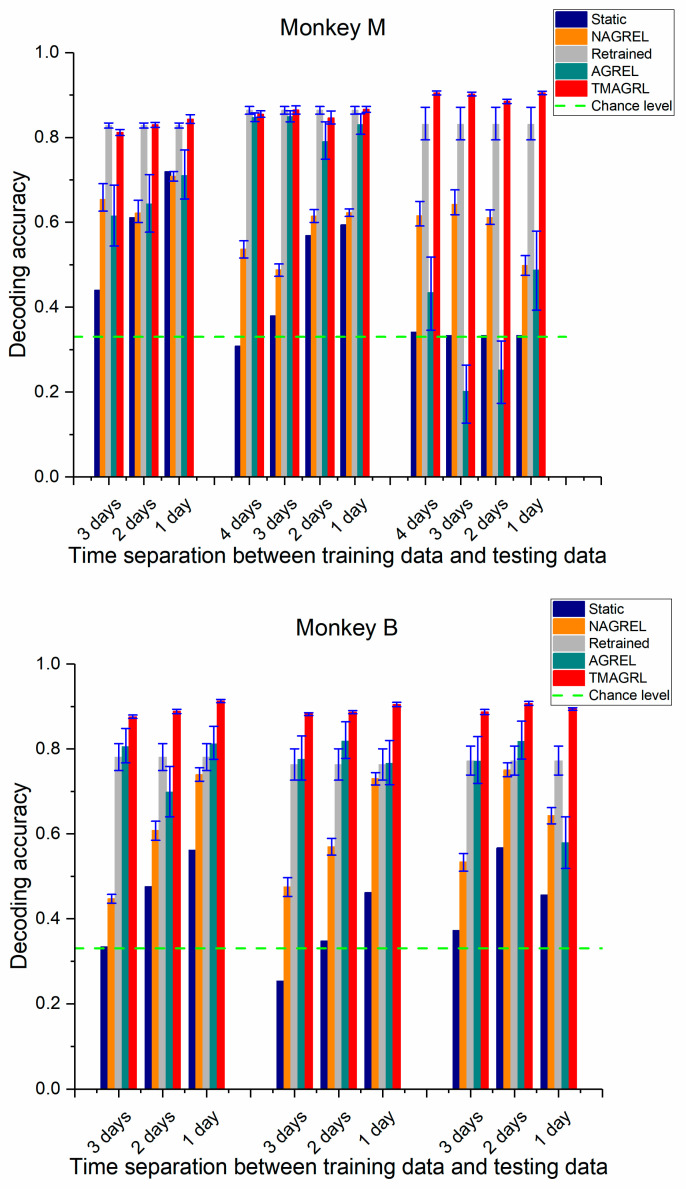
Performance comparison of the five methods across all sessions in both monkeys using historical data with a longer time separation. The decoding accuracy of each dataset was the mean value of 50 repeated computations. Blue, yellow, gray, green, and red bars represent the classification accuracy of the decoders calibrated using the static decoder, NAGREL, retrained decoder, AGREL, and the proposed TMAGRL, respectively. Error bars represent the confidence intervals of the mean decoding accuracies.

**Figure 8 sensors-20-05528-f008:**
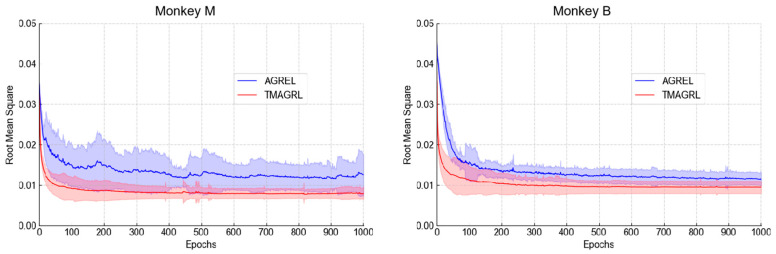
The root mean square value between the predicted and the actual values during the initial decoder training with dataset 1 of session one in monkeys M and B. The blue and red curves represent AGREL and the proposed TMAGRL, respectively. The procedure was repeated 50 times to achieve the mean values and standard deviations.

**Figure 9 sensors-20-05528-f009:**
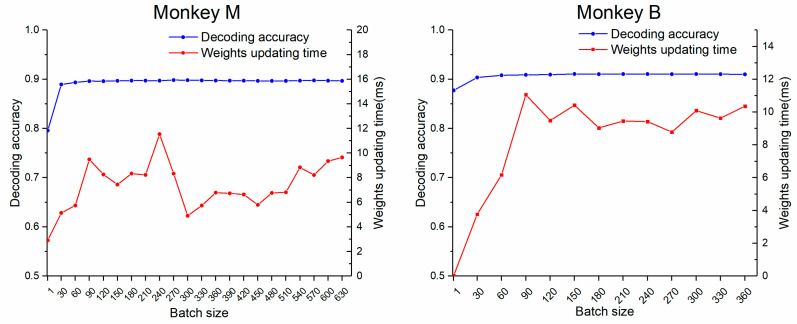
Effect of batch size on the decoding accuracy and the weight updating time in both monkeys. The blue and the red lines represent decoding accuracy and weight updating time under different batch sizes, respectively.

**Table 1 sensors-20-05528-t001:** Mean decoding accuracies and standard deviations for all data sessions of five methods based on historical data from the previous day.

**Monkey M**					
**Method**	**Static**	**NAGREL**	**Retrained**	**AGREL**	**TMAGRL**
Accuracy (%)	67.8 ± 14.4	75.9 ± 13.8	85.9 ± 4.2	73.9 ± 21.0	89.8 ± 4.2
**Monkey B**					
**Method**	**Static**	**NAGREL**	**Retrained**	**AGREL**	**TMAGRL**
Accuracy (%)	70.4 ± 17.7	79.5 ± 7.7	85.1 ± 6.2	81.7 ± 8.8	91.0 ± 2.3

AGREL: attention-gated reinforcement learning; NAGREL: non-adaptative AGREL; TMAGRL: transfer learning and mini-batch based attention-gated reinforcement learning.

**Table 2 sensors-20-05528-t002:** The confusion matrix of decoding results by TMAGRL for two monkeys.

**Monkey M**				
**Confusion Matrix**		**Predicted Label**		
		**left**	**middle**	**right**
Actual label	left	189	18	3
	middle	16	185	9
	right	4	14	192
**Monkey B**				
**Confusion Matrix**		**Predicted Label**		
		**cube**	**triangle**	**sphere**
Actual label	cube	108	5	7
	triangle	4	107	9
	sphere	5	3	112

**Table 3 sensors-20-05528-t003:** The mean decoding accuracies and standard deviations for all data sessions of five methods based on historical data with greater time separation.

**Monkey M**					
**Method**	**Static**	**NAGREL**	**Retrained**	**AGREL**	**TMAGRL**
Accuracy (%)	45.1 ± 14.5	60.1 ± 6.7	84.2 ± 1.7	60.5 ± 23.4	86.5 ± 3.2
**Monkey B**					
**Method**	**Static**	**NAGREL**	**Retrained**	**AGREL**	**TMAGRL**
Accuracy (%)	42.5 ± 10.6	61.1 ± 11.4	77.1 ± 0.8	76.0 ± 7.8	89.3 ± 1.3

**Table 4 sensors-20-05528-t004:** The time spent in parameter updating during the online testing of both monkeys.

**Monkey M**											
**Dataset Name**	**S1D2**	**S1D3**	**S1D4**	**S2D2**	**S2D3**	**S2D4**	**S2D5**	**S3D2**	**S3D3**	**S3D4**	**S3D5**
AGREL (ms)	126.5	311.9	246.6	26.7	142.6	124.3	387.1	839.0	752.2	536.3	804.1
TMAGRL (ms)	2.8	2.3	6.1	1.0	1.1	1.8	3.4	10.2	12.0	9.9	6.1
**Monkey B**											
**Dataset Name**	**S1D2**	**S1D3**	**S1D4**		**S2D2**	**S2D3**	**S2D4**		**S3D2**	**S3D3**	**S3D4**
AGREL (ms)	119.2	213.5	298.4		210.9	198.7	453.5		246.5	647.7	296.5
TMAGRL (ms)	1.0	5.3	2.9		0.2	0.7	9.1		1.7	5.6	8.1

S: session; D: dataset.

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
