# Peer review of "Reinforcement Learning Based Fast Self-Recalibrating Decoder for Intracortical Brain–Machine Interface"

_sensors, 2020, doi:10.3390/s20195528_

Round 1
Reviewer 1 Report
The authors proposed an attention-gated reinforcement learning based algorithm that combined transfer learning, mini-batch, and weight updating strategies to accelerate the weight updating and achieve fast self-recalibrating decoder for intracortical brain-machine interface. Experimental analysis was conducted on brain data collected from two monkeys using a 128-channel system. Though the presented study is interesting and meaningful for advancing the field, some critical comments need to address to improve the manuscript.
I would suggest providing some examples to visualize the feature representation learned by neural network. This can help readers better understand how the method works on feature mapping.
Some important information about the experimental study should be clarified. Please give more details about (1) how many samples were used for model training and how many for testing? (2) What type of cross-validation was implemented to evaluate the experimental performance? I would suggest using a K-fold (e.g., 10-fold) cross-validation.
Regarding classification performance, the authors presented the overall accuracy. I would also like to see a confusion matrix that shows the accuracy of classify each of the action states.
Some closely related deep learning methods have been recently developed for brain signal analysis. The authors may review more of these studies, such as: A survey on deep learning based brain computer interface: Recent advances and new frontiers; Adversarial representation learning for robust patient-independent eplileptic seizure detection.
The authors may briefly discuss the potential limitations of the proposed method and what are the future research directions of this study. How other researchers can work on your study to continue this line of research?
Reviewer 2 Report
Abstract
Line 19: don’t say “can be solved”, - “can be improved/enhanced” would be better
Line 21: pluralize decoder – decoders.
Introduction
A brief explanation of RL may be useful. The general concepts of how a RL system operates would provide context for some readers
Line 37: “… record neural activity which is then…”
Line 39: remove “a” and “the”
Line 40: replace “allow” with “assist”; insert “while” between limbs and performing
Line 42: It would be helpful to state than when making reference to the “decoder”, you are referring to a general set of algorithms commonly used in BMI decoding.
Line 51: Change to, “…studies have implemented this method…”
Line 56: No need to re-state reinforcement learning (RL)
Line 58: It would be useful to readers if a definition is provided for the term “environment”. Environment can have a particular meaning in RL but MBI reader might not be familiar with it.
Line 86,87: I would not say that “knowledge in the historical data can be transferred to the new data” – it would be more accurate to say that knowledge extracted from the historical data can be used to prime the RL decoder with relevant information.
Line 89: should be, “…may be due to the fact that…”
Line 96: State the full name before TMAGRL
Is the approach faster than previous methods with time required to implement transfer learning included? Or is it faster after TL has already been applied?
Line 100: should “mode” be “model”?
Line 101: may be better to state that the approach helped to mitigate against overfitting rather than completely avoided it.
Materials and Methods
Figure 1(c) would benefit from inclusion of the approximate timings.
Was there a specific reason why the number of electrode arrays were different between the two monkeys.
Line 162: remove “the”
Section 2.2 – it would be useful to include a description of how the spikes (what was the method) were extracted from the data.
It is not clear to me why different lengths of time pre- and post-target hit event were selected. Is it because of some difference between shape and target objects?
Line 187: State the full name of the algorithm here
When referring to historical data, please state precisely which data this is. For example, does historical data refer to sessions 1 and 2, current data session 3?
Line 307: To be clear, were methods 3 and 4 trained using the same data split as method 1? i.e. historical data for training and current data for testing?
What is the architecture of the static decoder? It is not clear.
Results
Line 329: Monkey identifiers are misaligned in Table 1
Figure 6 us a little squeezed and difficult to read. It might be better to vertically stack the plots.
Line 341: The specific type of ANOVA test should be stated and the full result reported. For example, F(2,250)=10, p<0.5
Paragraph beginning line 336: Some of the information presented here might be better places in the discussion, particularly parts in which you discuss why one algorithm performs better in comparison with another.
Line 361: This line suggests that separate results for tests on data with different number of days separations were compared. However, the presentation of these results in Figure 7 is difficult to read. It might be useful to organize these results into a table instead of presenting the mean results as in Table 2.
Paragraph beginning line 387 may be better placed in the methodology section.
Figure 8: label the plots for Monkey B and M.
Table 3: reformat the table to similar style as previous tables
Discussion
The discussion section could benefit from more examples in the literature of approaches used to mitigate the effects of EEG non-stationarity across multiple sections. It would be interesting to know more about how this RL method compares to other non-RL methods that have been applies
Reviewer 3 Report
The manuscript uses reinforcement learning with intracortical brain-machine interface data. It is well organized and well written, with adequate figures. As this first version is already in a good shape, I have only a few suggestions and general comments below.
In deep learning techniques, ‘decoder’ is commonly used to name a part of the model architecture, usually the part of the model closer to the output. I was a little unsure if ‘decoder’ is used in the deep learning sense or if it is used to reference part of the iBMI at the beginning of the manuscript. I later concluded it refers to the model, but I suggest adding a short definition of ‘decoder’ soon into the paper to help the reader.
The authors use the concept of online testing and mini-batch sample sets. I suggest adding some comments and discussion on the relationship of such approach with the concept of experience replay and prioritized experience replay (e.g., https://arxiv.org/abs/1511.05952). Mini-batch seems to be one way of doing experience replay.
Minor comments
Although I’ve seen section division in the abstract (Background, Methods, Results, Conclusions) in other journals, I don’t think that is the standard for Sensors. Maybe you can consider removing them or checking with the editor.
Line 30: RLBMI was not defined in the abstract.
Figure 5: the word ‘Convergence’ is split. I believe this is purely an aesthetics choice, but I suggest increasing the boxes to make sure the words fit.
Round 2
Reviewer 1 Report
The authors have addressed my concerns. The revision is looking acceptable for publication.